# Catching the Wave: Detecting Strain-Specific SARS-CoV-2 Peptides in Clinical Samples Collected during Infection Waves from Diverse Geographical Locations

**DOI:** 10.3390/v14102205

**Published:** 2022-10-07

**Authors:** Subina Mehta, Valdemir M. Carvalho, Andrew T. Rajczewski, Olivier Pible, Björn A. Grüning, James E. Johnson, Reid Wagner, Jean Armengaud, Timothy J. Griffin, Pratik D. Jagtap

**Affiliations:** 1Department of Biochemistry, Molecular Biology, and Biophysics, University of Minnesota, Minneapolis, MN 55455, USA; 2Division of Research and Development, Fleury Group, São Paulo 04344-070, Brazil; 3Département Médicaments et Technologies pour la Santé (DMTS), Université Paris-Saclay, CEA, INRAE, 30200 Bagnols-sur-Cèze, France; 4Department of Computer Science, University of Freiburg, 79110 Freiburg, Germany; 5Minnesota Supercomputing Institute, University of Minnesota, Minneapolis, MN 55455, USA

**Keywords:** SARS-CoV-2, variant detection, strain-specific, mass-spectrometry

## Abstract

The Coronavirus disease 2019 (COVID-19) pandemic caused by the severe acute respiratory syndrome coronavirus 2 (SARS-CoV-2) resulted in a major health crisis worldwide with its continuously emerging new strains, resulting in new viral variants that drive “waves” of infection. PCR or antigen detection assays have been routinely used to detect clinical infections; however, the emergence of these newer strains has presented challenges in detection. One of the alternatives has been to detect and characterize variant-specific peptide sequences from viral proteins using mass spectrometry (MS)-based methods. MS methods can potentially help in both diagnostics and vaccine development by understanding the dynamic changes in the viral proteome associated with specific strains and infection waves. In this study, we developed an accessible, flexible, and shareable bioinformatics workflow that was implemented in the Galaxy Platform to detect variant-specific peptide sequences from MS data derived from the clinical samples. We demonstrated the utility of the workflow by characterizing published clinical data from across the world during various pandemic waves. Our analysis identified six SARS-CoV-2 variant-specific peptides suitable for confident detection by MS in commonly collected clinical samples.

## 1. Introduction

It has been more than two years since the Coronavirus disease 2019 (COVID-19) outbreak, which has since spread worldwide, resulting in almost 6.3 M deaths [1]. Infected patients have exhibited symptoms that range from asymptomatic to mild fever, cough, and myalgia to severe respiratory distress, organ failure, and death in critical cases. Methods for the detection of viral infection, as well as vaccines and therapeutics, have improved the situation; however, continuous viral mutations, especially in low-vaccinated geographical regions [2], have led to the emergence of new variants at different locations and times across the world. These variants show distinct characteristics with respect to incubation times, infection routes, and severity of the disease [3]. This has posed serious challenges at multiple levels including a) medical intervention by healthcare workers [4]; b) detection of new strains carrying new genetic mutations in the population by clinical labs; c) vaccine efficacy for pharmaceutical companies [5,6], and d) monitoring of the temporal and geographical course of the pandemic for the scientific community [7]. It became extremely critical to detect the new strains using molecular techniques to monitor the progression of new waves of the pandemic. The characterization of the viral protein sequences specific to newly defined variants is important, as it directly identifies the structural molecules (nucleocapsid, membrane, and spike proteins) that are recognized by antigen tests and are important molecular targets for vaccine development and other therapeutics [8].

Most commonly, severe acute respiratory syndrome coronavirus 2 (SARS-CoV-2) infection is detected using the RT-PCR of patient-derived swab samples, or at-home kits recognizing viral antigens [9]. Despite the effectiveness of these diagnostic tests for the rapid detection of viral infection, other approaches offer a more in-depth characterization of clinical samples [10,11]. Mass spectrometry (MS)-based proteomics provide an orthogonal method to understand the status of infection by directly characterizing the virus-expressed proteins from commonly collected clinical samples (e.g., nasal swabs) [12]. MS-based instrumentation platforms can characterize many clinical samples with high throughput. For example, a turbulent flow chromatography-mass spectrometry (TFC-MS) system method developed by Carvalho and colleagues allows a high-throughput multiplexed analysis of more than 500 samples per day [13]. A customized bioinformatics analysis that allows for the biological interpretation of the complex MS data that are generated is also a requirement to understand the dynamics of viral-protein expression from clinical samples. Fortunately, these bioinformatic tools exist. In our previous studies, we published MS-proteomics-based informatics workflows that can detect and verify SARS-CoV-2 peptides, including those specific to characterized variants [14] and also peptides from potential co-infection pathogens [15] that may be specific to infection waves. These are deployed within the Galaxy bioinformatics ecosystem [16], providing a workbench wherein scientists can share, analyze, and visualize their results in a reproducible manner, carrying out analyses on scalable computer resources accessed through a web browser interface. The ecosystem also provides extensive online and on-demand training material via the Galaxy Training Network [17], including guidance on the usage of the platform for SARS-CoV-2 studies [18]. 

In this study, our MS proteomics-based Galaxy workflows identified six SARS-CoV-2 variant-specific peptides from published clinical samples. This was achieved by extending our previously published workflows by analyzing 12 previously generated and published MS-based proteomics datasets from a variety of geographical areas and infection wave timelines. These datasets cover a timeline from March 2020 to January 2022 and cover seven countries and three continents (Figure 1, Appendix A). We leveraged the flexibility of workflows in Galaxy to match the peptide mass spectra acquired by tandem mass spectrometry (MS/MS) within these published datasets and against an updated SARS-CoV-2 protein sequence database, including sequences specific to variants classified by the World Health Organization (WHO) [19]. Our discovery workflow first detects SARS-CoV-2 peptides of specific sequences from the clinical samples. Identified peptides from all the datasets are combined and re-evaluated using the PepQuery [20] search engine to verify the presence of these peptide spectrum matches (PSMs) before validating their spectral quality by visual inspection. After the diligent evaluation and confirmation of spectral quality, the resultant peptides are aligned against the wild-type SARS-CoV-2 proteome to identify peptide sequences specific to WHO-classified variants and their associated phylogenetic lineages. Our results from the re-analysis of these datasets provided a demonstration of the power of this approach, characterizing protein sequence changes and verifying MS-detectable peptides that follow temporal and geographic dynamics of SARS-CoV-2 infection waves from diverse clinical sample types. In addition, our bioinformatics tools and workflows that generate these results are well-documented and freely and easily accessed, providing a means for others to utilize this approach in the characterization of SARS-CoV-2 samples or potential studies of other viral infections. 

## 2. Materials and Methods

### 2.1. Clinical Datasets

Twelve clinical MS datasets (Figure 1) available via the ProteomeXchange consortium were used to detect variant peptides and proteins. For our timeline and geographical evaluation of the COVID-19 infection, we chose five nasopharyngeal swab datasets (PXD019686 [21], PXD023016 [22], PXD034582 (August, September, and January)); a gargling sample dataset (PXD019423) [23]; a nasal swab (PXD020394); an upper respiratory tract sample (PXD021328) [13]; a BALF sample (PXD022085) and a urine sample (PXD024967) [24], all collected at different times and locations. Note that the datasets collected in Sao Paulo (PXD021328 and PXD034583) were pooled clinical samples, wherein each raw file contained data from two patients. Appendix A also provides more details on these previously published datasets.

### 2.2. Discovery Workflow

Our previous study [14] used two workflows: a) discovery workflow for COVID-19 peptide detection and b) verification workflow for confirmation using the PepQuery tool (Galaxy Version 1.6.2+galaxy1) [20]. The discovery workflow (Figure 2A) used several sequence database search algorithms, such as X! tandem, MSGF+, and OMSSA within SearchGUI (Galaxy Version 3.3.10.1) [25]/PeptideShaker (Galaxy Version 1.16.36.3) and Andromeda [26] within MaxQuant (Galaxy Version 1.6.17.0+galaxy3) [27] to detect PSMs, peptides, and infer proteins at a 1% global False Discovery Rate (FDR). The COVID-19 protein sequence database used for matching peptide MS/MS data consisted of the nucleocapsid, membrane, and spike protein mutations from the variants of concern (B.1.1.7, B.1.351, P.1, B.1.671.2, AY.4, AY.4.2, XE, B.1.1.529, BA.1, BA.2, BA.3, BA.4). Along with variant structural proteins, we added all the proteins from the wild-type strain (EPI_ISL_402124—dated 30 December 2019), and sequences were obtained from the GISAID database (https://www.gisaid.org/, last accessed on 3 February 2022). As the datasets were from clinical samples, we added human proteins and common contaminants to the COVID-19 protein sequence database.

The sequence database search parameters used for digestion, modifications, tolerance, and FDR estimation were consistent with the parameters mentioned in the published papers for each of these datasets (Appendix A). Confident SARS-CoV-2 peptides from all the datasets were parsed out by eliminating the human and the common contaminant peptides. The Galaxy-based discovery workflow can be accessed here (https://usegalaxy.eu/u/galaxyp/w/coviddiscovery-workflow; last accessed 21 September 2022).

### 2.3. Peptide Verification Workflow

We detected 203 SARS-CoV-2 datasets (Appendix A). peptides from the 12 clinical MS datasets using our discovery workflow and customized protein sequence database. The 203 detected peptides were compared to the existing 803 SARS-CoV-2 peptides from previously published data [14,28], resulting in 103 unique peptides for this current analysis (Appendix A). These peptides were then combined to create a larger SARS-CoV-2 peptide panel. The combined peptide panel of 906 peptides was then subjected to the verification workflow (Figure 2B) to confirm the veracity of the PSMs identifying these peptides, as well as the spectral quality. The peptide verification workflow performed a re-analysis of the datasets using the PepQuery tool parameters specified in (Appendix A). PepQuery filters out putative SARS-CoV-2 PSMs that may be better matches to human or contaminant protein sequences that are present in the background reference database, as well as further confirming those PSMs that best match viral proteins. Confident viral peptides with a *p*-value ≤ 0.05 assigned by PepQuery were then subjected to spectral visualization and manual inspection using the Multiomics Visualization Platform (MVP) tool [29] within the Galaxy platform to ascertain the quality of verified peptides (Appendix A). As the peptide spectral quality is crucial for developing reliable targeted MS-based assays, we further validated the PepQuery-filtered peptides using criteria that included each spectrum containing three consecutive b- and/or y-ions detected, and the MS2 ion intensities considered were at least three-fold higher than the background noise (Appendix A). The peptides that passed the manual inspection were then subjected to BLAST-P analysis against the Wild-Type SARS-CoV-2 proteins and the non-redundant database (NCBI-nr). The wild-type SARS-CoV-2 proteome sequence was obtained from the GISAID database and represents the reference virus strain characterized at the earliest stages of the pandemic (EPI_ISL_402124—dated 30 December 2019). Additionally, the annotated viral peptides were reconfirmed as belonging to SARS-CoV-2 variants using the peptide reports from Search GUI/PeptideShaker (Peptide Report) and MaxQuant (peptides.txt) from the initial discovery analysis. This comparison along with BLAST-P analysis showed that there were 17 peptides unique to viral strains. 

To further annotate these sequences, a Phylogenetic Assignment of Named Global Outbreak (Pango) lineage analysis; last accessed on 3 February 2022 [30] was performed using a python script. For the Pango lineage search, a GISAID protein FASTA file (allprot0203) and an associated metadata file were downloaded from https://www.gisaid.org/; last accessed on 3 February 2022. The allprot0203.fasta file included 205,705,355 sequences for a size of 113,874,658 KB. The metadata.tsv file included 7,786,913 accession IDs for a size of 4,586,043 KB. The list of 17 peptide sequences described previously was used to subset the allprot0203.fasta file by keeping only proteins containing at least one of the peptides, with I/L residues undifferentiated. An in-house python script was used on this file for (i) the in-silico digestion of these proteins with two missed cleavages allowed; (ii) mapping of all protein sequences to each peptide from the list of 17 sequences; (iii) retrieval of all matching information (including the Pango Lineage) from the metadata file through the GISAID Accession ID; and (iv) summary of information per peptide using the “groupby” function from the python pandas package. Retained information includes (Appendix A) the list of unique proteins, a list of Pango lineages, the first five and last five countries of appearance, the first GISAID Accession ID, and the number of associated GISAID Accession IDs. For additional verification, the peptides assigned to mutated sequences were manually aligned to the wild-type strain to verify their specificity to defined viral variants. The variants were classified according to the WHO-SARS naming system [19], and the amino acid sequence specific to these variants was confirmed by referring to the current SARS-CoV-2 lineage database [31]. The peptide verification workflow can be accessed here (https://usegalaxy.eu/u/galaxyp/w/covid-verification-workflow; last accessed 21 September 2022). 

## 3. Results

### 3.1. Discovery Workflow Results

The discovery workflow performed sequence database searching using two software platforms (SearchGUI-PeptideShaker and MaxQuant). The sequence database searching of the clinical MS datasets against the database of protein sequences specific to SARS-CoV-2 variants confidently detected 203 unique peptides (Appendix A). These peptides mainly represented structural proteins from the SARS-CoV-2 proteome, mostly from the Nucleocapsid protein. In our previous published study, a panel of 623 peptides was generated from MS data generated from cell culture and clinical samples, as well as a study using in silico-translated viral protein sequences, to generate PSMs and identify viral peptides [21,28]. We merged this list of peptides with an additional list of peptides from a study employing spectral-library searching against the wild-type SARS-CoV-2 proteome [32] to generate a list of 803 peptides. We found an overlap of 100 peptides after comparing the 203 detected peptides from the discovery workflow in our current study with 803 peptides from previous studies. All the high confidence SARS-CoV-2 peptides from our current and previous studies detected by MS/MS were merged, resulting in a total of 906 peptides.

### 3.2. Verification Workflow Results

The comprehensive panel of 906 peptides was used to re-interrogate the 12 clinical MS datasets using the PepQuery tool within our verification workflow. PepQuery stringently evaluates the veracity of these putative virus-specific PSMs by the re-analysis of corresponding MS/MS spectra against the human and common contaminant protein sequences, including possible post-translational modifications [20]. Those MS/MS spectra that still best match their viral peptide sequence after this analysis are passed on for further consideration. Out of the 906 peptides, 82 high-quality peptides (Appendix A) passed the PepQuery confidence filter (*p*-value ≤ 0.05) and the subsequent manual spectral quality inspection using Lorikeet (Figure 3). The manual annotation and Blast-P results showed that 75.6% of these high-quality peptides were from the Nucleocapsid, 17.1% from the Spike protein, 4.9% from Membrane proteins, and 2.4% from the NS9b protein.

Blast-P analysis along with Pango lineage analysis [30] assigned the peptides to proteins that may be specific to SARS-CoV-2 variants. The peptides in Table 1 and Appendix A show that the verified variant peptides belonged to the nucleocapsid protein. They were further annotated using WHO nomenclature [19] and the associated Pango lineage.

To assess the specificity of these peptide sequences to viral variants, we also manually aligned them to the wild-type sequences to confirm their identities (Table 1 and Figure 3). As a result of this evaluation, we identified six peptides that belong to the variants Gamma, Delta, and Omicron. Despite the nucleocapsid being the most invariant protein sequence in SARS-CoV-2 [33], we observed sequence changes in nucleocapsid peptides specific to variants, such as P80R in Gamma, D377Y, R203M and A208T in Delta [31], and the deletion of amino acid positions 31–33 in Omicron [34]. All of the mentioned peptides were present in clinical samples obtained from COVID-19-positive patients collected at different times and geographical locations; hence, along with their rigorously verified identities from MS/MS spectral analysis, these peptides deserve consideration as optimal candidates for detection using targeted MS-proteomics (selected reaction monitoring (SRM) and parallel reaction monitoring (PRM)) [35] or potentially other detection platforms.

## 4. Discussion and Conclusion

The different COVID-19 pandemic waves have been driven by constantly evolving virus strains that vary in their virulence, fatality, severity, and infectivity [36]. These waves have also been dynamic in the geographically affected areas and timepoints and have put strain on the healthcare system when they appear within populations. Scientists, from basic researchers to clinicians to epidemiologists, continue to monitor these emerging strains and classify them according to the WHO-SARS-CoV-2 naming system, which includes variants of concern, interest, to be monitored, or high consequence [19], depending on their contagiousness, clinical presentation, severity, and responsiveness to vaccines and/or therapies. Pango lineage nomenclature is a common way to classify distinct lineages of SARS-CoV-2 compared to the reference sequence [30]. Essential to these ongoing monitoring efforts is the ability to detect evolving sequence changes to the essential proteins of SARS-CoV-2 variants, which are the targets of rapid tests and, in some cases, vaccines and therapies. MS-based proteomics, in particular targeted methods against variant-specific peptides, offer a useful approach for such monitoring, directly characterizing these sequences from sample types commonly collected in the clinic. These methods, however, depend on verified peptide sequences specific to these variants that can serve as targets.

In this study, we presented two workflows, available within the Galaxy platform, which can identify and verify SARS-CoV-2 variant-specific peptides. These flexible workflows have the potential to detect new sequences from emerging strains on any clinical datasets analyzed using contemporary MS-based proteomics methods (e.g., data-dependent acquisition of MS/MS spectra, or even targeted parallel reaction monitoring MS/MS data [35]). These workflows are also publicly available, along with documentation, through the European-based Galaxy ecosystem [37]. We anticipate these being useful to a wide variety of researchers as COVID-19 research continues. Importantly, these workflows are composed of verification steps for peptides initially discovered by sequence database searching of MS/MS data, ensuring only verified and validated peptides are reported.

We continue to expand our panel to include peptide targets specific to emerging variants and their associated strain lineages. Our current panel contains 906 peptides shared across all strains of SARS-CoV-2, in which six variant-specific peptides useful for monitoring the infection dynamics of clinically distinct forms of the virus were characterized in this work. These peptides align to the nucleocapsid viral particles, which play critical roles in host infection mechanisms, as targets of antigen-based rapid tests, and could also help in therapeutic approaches [8,38,39]. These verified peptides are optimal for targeted MS-based proteomic assays that have been described for SARS-CoV-2 diagnostics and monitoring [13]. Given the growing emphasis on testing wastewater samples for COVID-19 surveillance [40], we also envisioned the potential use of these peptides as targets for high-sensitivity MS-based assays analyzing these emerging sample types. Such protein-based assays in environmental samples could provide an improved way forward for monitoring community infection dynamics. Although best-suited for diagnostic applications, direct monitoring of proteins specific to virus variants in communities and populations could help in the development and/or choice of the best targets for vaccines and even the development of other therapies aimed at minimizing the severe effects of infection.

In summary, we demonstrated the use of customized bioinformatic workflows to identify six confident SARS-CoV-2 variant-specific peptides suitable for MS/MS based detection in clinical samples. These peptides should be useful for developing targeted MS-based assays for the rapid and sensitive characterization of variant-specific proteins within clinical samples. Our discovery and verification workflows were developed within Galaxy and are accessible via the public and freely available European Galaxy resource (usegalaxy.eu; last accessed 21 September 2022), and as individual tools available in the Galaxy Tool Shed [41]. These workflows are flexible, with amenability to further customization for individual datasets. Although our focus has been SARS-CoV-2 characterization, our workflows could be adapted to MS-based proteomics data from other pathogenic organisms, providing value to the broader infectious disease research community.

## Figures and Tables

**Figure 1 viruses-14-02205-f001:**
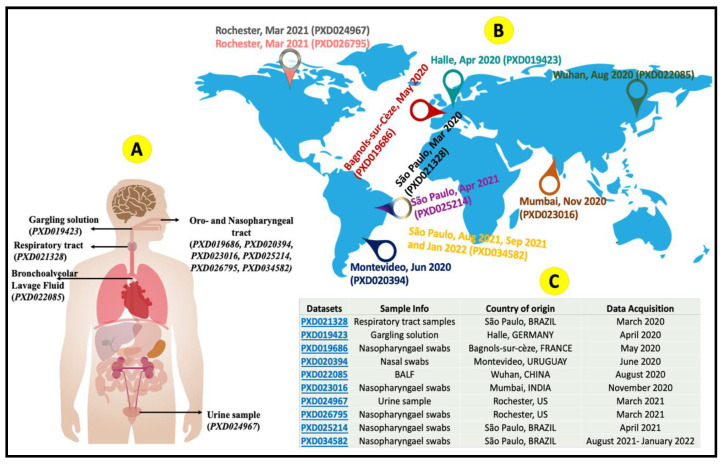
Publicly available clinical tandem mass spectrometry (MS/MS) datasets from ProteomeXchange were used for the variant detection study. (**A**) Samples came from different parts of the human body. (**B**) Samples and generated datasets were obtained at different timepoints and locations, following the geographical and temporal dynamics of the infection waves. (**C**) Table summarizing the ProteomeXchange accession numbers and geographical and temporal information associated with each dataset.

**Figure 2 viruses-14-02205-f002:**
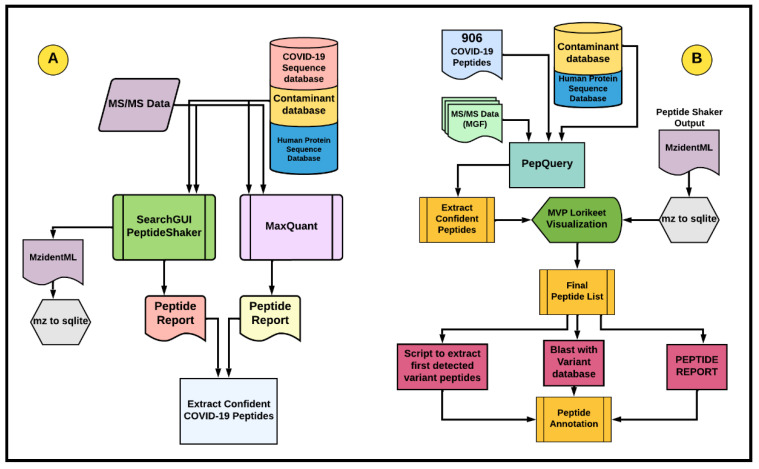
Galaxy-based workflows were used to identify and verify SARS-CoV-2 peptides from MS-based clinical datasets. (**A**) **Discovery workflow:** MS/MS spectra from clinical datasets were searched against a database consisting of SARS-CoV-2 structural protein sequences from SARS-CoV-2 variants, wild-type virus protein sequence, common contaminant sequences, and human protein sequences using SearchGUI/Peptide Shaker and MaxQuant. The PSM output was filtered to extract confident matches to COVID-19 peptides with sequences unique to viral variants. mzidentML generated from SearchGUI-Peptide shaker was used for spectral quality analysis via Lorikeet viewer. (**B**) **Verification Workflow:** A peptide panel of 906 SARS-CoV-2 peptides (theoretical and empirically detected peptides obtained from in silico analysis, cell-culture, and clinical datasets including variant-specific peptide sequences) was subjected to the PepQuery analysis of clinical MS datasets. The quality of the verified PSMs was manually interrogated using the Lorikeet visualization platform within the Multi-omics Visualization Platform (MVP) for additional evaluation. High-quality, confident peptides were confirmed for specificity to virus variants by Blast-P analysis and were annotated for associated phylogenetic lineages by Pango lineage analysis. Finally, these annotated peptides were compared to the original PeptideShaker (peptide report) and MaxQuant (peptide.txt) discovery results to confirm their initial matches to the specific protein sequences and also that they belonged to these virus variants.

**Figure 3 viruses-14-02205-f003:**
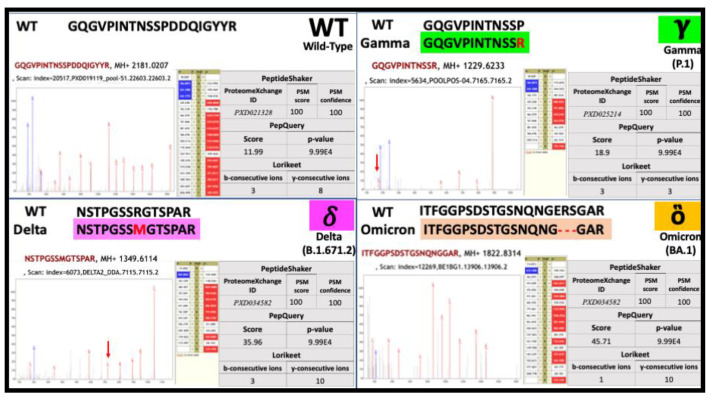
Representative MS/MS spectra of SARS-CoV-2 verified peptides from the clinical samples. The figure shows the manually validated and annotated MS/MS spectra resulting in these peptide sequence identifications. Representative variant-specific peptide sequences from the Nucleocapsid protein are shown, along with their alignment to the wild-type sequences and scores from the discovery, verification, and validation steps. The peptides in black font are from wild-type SARS-CoV-2, while the variant peptides are colored-coded; assignments to WHO-designated variants are shown as Gamma (green), Delta (pink), and Omicron (orange). All the amino acid sequence mutations are marked with red text, and the red arrow within the annotated MS/MS spectra designates the detection of sequence fragments at m/z values specific to fragments carrying these mutations.

**Table 1 viruses-14-02205-t001:** Verified peptides with variant-specific sequences to the SARS-CoV-2 nucleocapsid protein.

PEPTIDE	WT SEQUENCE	Variant (WHO Name)	BLASTP IDENTITY (WT)	BLASTP IDENTITY (NR)
GQGVPINTNSSR (P80R)	GQGVPINTNSSP	P.1.(Gamma)	88.00	100.00
AYETQALPQR(D377Y)	ADETQALPQR	B.1.617.2 (Delta)	80.00	100.00
GEGVPINTNSSPDDQIGYYR (Q69E)	GQGVPINTNSSPDDQIGYYR	B.1.1.7 (Delta)	95.00	100.00
SMGTSPTRMAGNGGDAALALLLLDR (R203M & A208T)	SRGTSPARMAGNGGDAALALLLLDR	B.1.617.2 (Delta)	86.67	96.00
PGNGCDAALALLLLDR (A211P &G215C)	AGNGGDAALALLLLDR	AY.4 (Delta)	93.33	100.00
ITFGGPSDSTGSNQNGG|AR (*∆*31–33)	ITFGGPSDSTGSNQNGERSGAR	BA.1 (Omicron)	86.36	100.00

The table shows different nucleocapsid peptides with sequences specific to virus variants (shown in parentheses), the wild-type sequence, and their assigned Pango lineage identifier. Mutated amino acids are shown in the red text along with the amino acid changes at their specific positions in the primary protein sequence. The Blast-P similarity is shown in both wild-type SARS-CoV-2 sequences and the Non-Redundant (NR) NCBI database. The NCBI NR-database contains many of the variant-specific sequences; thus, many of these showed 100% similarity.

## Data Availability

The data have been made available via the Zenodo platform- https://doi.org/10.5281/zenodo.7153337 accessed on 4 October 2022. The datasets were downloaded according to their PXD identifier from ProteomeXchange using url: http://proteomecentral.proteomexchange.org/cgi/GetDataset; last accessed 4 October 2022. The Galaxy workflows along with a sample input data and results are available via https://covid19.galaxyproject.org/proteomics/; last accessed 21 September 2022 and the Galaxy HUB—https://galaxyproject.org/community-projects/covid-proteomics/; last accessed 4 October 2022.

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
