# Peer review of "Catching the Wave: Detecting Strain-Specific SARS-CoV-2 Peptides in Clinical Samples Collected during Infection Waves from Diverse Geographical Locations"

_viruses, 2022, doi:10.3390/v14102205_

Round 1

Reviewer 1 Report

In the submitted manuscript entitled "Catching the Wave: Detecting strain-specific SARS-CoV-2 peptides in clinical samples collected during infection waves from diverse geographical locations" authors described bioinformatics tools and workflows that enable the characterization of peptides specific to the different SARS- CoV-2 variants. This study presents an upgrade of the authors previous study and highlights the utility of using MS-based methods to monitor the status of the SARS-CoV-2 pandemic.

Described methods and results are well presented and supported with supplementary material with some minor issues (ex. In Figure 1 PXD022085 is mentioned as a BALF sample and also as an Oral and Nasopharyngeal tract sample).

It is not clear why in the manuscript title is highlighted that samples originate from different locations due to the global spreading of SARS-CoV-2 variants and strains. Furthermore, I suggest reconsidering some references (ex. reference 3 is not describing distinct characteristics of the SARS-CoV-2 variant with respect to incubation times, infection routes, and severity of the disease).

In general, the submitted manuscript clearly presented bioinformatics tools and workflows for variant specific peptides characterization which could be used in the further monitoring of the SARS-CoV-2 pandemic. 

Author Response

We would like to thank the reviewer for their comments. The responses are given in green.

POINT 1: Described methods and results are well presented and supported with supplementary material with some minor issues (ex. In Figure 1 PXD022085 is mentioned as a BALF sample and also as an Oral and Nasopharyngeal tract sample).

RESPONSE 1: Thank you for the reviewer’s comment. We realize the minor issue with the PXD022085 dataset and have made the required changes. We have removed PXD022085 from the Oro-Nasopharyngeal tract in Figure 1.

POINT 2: It is not clear why in the manuscript title is highlighted that samples originate from different locations due to the global spreading of SARS-CoV-2 variants and strains. Furthermore, I suggest reconsidering some references (ex. reference 3 is not describing distinct characteristics of the SARS-CoV-2 variant with respect to incubation times, infection routes, and severity of the disease).

RESPONSE 2:Thank you for the reviewer’s comments. We would like to mention that the results generated using our workflows demonstrate how we can detect the peptide sequences specific to these virus variants. This correlated with geographical and temporal aspects of the spread of the virus across the globe, which has been a key feature of the pandemic.  Therefore we emphasized the ability to detect and correlate viral variants to geographical locations in the title as an important aspect of our work.  We mention the pandemic's geographic and temporal features in the Introduction (e.g. see lines 40-52 in the revised manuscript). We have also replaced reference 3 with an appropriate reference (i.e.Wu Y, Kang L, Guo Z, Liu J, Liu M, Liang W. Incubation Period of COVID-19 Caused by Unique SARS-CoV-2 Strains: A Systematic Review and Meta-analysis. JAMA Netw Open. 2022;5(8):e2228008. doi:10.1001/jamanetworkopen.2022.28008).

Reviewer 2 Report

Mehta et al. try to establish a bioinformatics workflow to identify strain specific SARS-CoV-2 peptides suitable for MS detection. To accomplish this goal the authors have re-analyzed 12 datasets from different time points and geographical locations. After identifying 103 previously undetected peptides, the authors reevaluated the 12 datasets with the now total 906 detected peptides through a second bioinformatics workflow mainly based on PepQuery and manual confirmation. The authors were able to identify six strain specific peptides corresponding to different Delta, Gamma and Omicron sub-strains. Overall, the manuscript is of interest but describes a very specific field of application, retrospectively identifying variant specific peptides. Thus, samples could later on be assigned permitting sufficient quantity. The manuscript in itself is coherent but very hard to read and in part lacks clarity. Therefore, the manuscript may eventually be publishable in Viruses after major revision.

Major comments:

-How and where data was generated and then used further during the workflows is not always easy to follow. Some details are missing and some references are not the primary literature while at least one simply does not contain the implied statement, e.g. reference 2. Thus, the manuscript requires sharpening and some rewriting.

-The authors tried to establish two bioinformatics workflows to identify strain-specific MS-detectable SARS-CoV-2 peptides. From a list of 906 unique peptides, six strain specific, high quality peptides were identified. While in general this workflow can be useful to identify MS-detectable peptides after initial analysis, it is still very limited to the knowledge of a specific strain. Furthermore, the authors claim usefulness for monitoring, vaccine and therapeutics development, without discussing any of the limitations of MS with respect to existing techniques. The set goal of reliably identifying strain specific peptides was accomplished but should be discussed as what it is, the identification of strain-specific peptides using MS (see also below).

-l. 80 Which sequences were used and where were they taken from?

-l. 103 How were these datasets selected? This is crucial, since the detected peptides depend on this, in the first as well as the second round. Along those lines, in l. 105-108 why are some datasets listed but others were not? In l. 105-108, more background for these sets are needed (single samples pooled?).

-l. 167 Define peptide reports used, MQ does not generate a peptide report.

-l. 173 List of 17 previously described peptides. Described where?

-l. 199 “in silico translated protein sequences” – overall this makes sense but in this context a bit confusing, since the work is based on empirical data. Please clarify.

-l. 216 Define quality (in spectral quality), how was that assessed. It would be useful and supportive to add additional example spectra (rejected, included/assigned) to the supplement. See also Fig. 3 - Omicron: here the 3x3 rule is not met since only one consecutive b-ion despite many y-ions. This seems to be a high confidence peptide. Please clarify when to break the 3x3 rule.

-l. 273-276 The existing datasets were processed using a defined sequence. In this case it is not possible to define new, previously unknown peptides.

-l. 295-300 Monitoring peptides directly is desirable, however the usefulness for vaccine and therapeutic development remains limited. The limitations of the approach need to be discussed. For example: MS can (with little exceptions) only detect structural viral proteins. No changes in the polymerase or nsp3, which are targets for antivirals and the latter has an elevated mutation rate compared to other non-structural proteins, could be detected.

Minor

-Throughout the text some words were used very repetitively mainly rigorous and rigorously. On most occasions, this word has no added value and could be removed.

-l. 15-16 Repetition of caused

-l. 38 Repetition of severe

-l. 41 wrong source

-l. 43 not the primary source

-l. 78 “cover major continents” has no meaning, just 4 continents were covered. Why are certain continents considered major and others minor.

-Fig. 1 - PXD020394 was a nasal swab not a nasopharyngeal swab according to ProteomeXchange and PXD034582 is not available yet

-Table 1 First line very thick use the width of the page

-l. 240 Seems redundant

-l. 263 Period missing

-l. 284 It is not clear if it is 906 peptides including strain specific or 912 peptides. Rephrase.

-l. 302 change is to in

Author Response

We would like to extremely thank the reviewer for their comments. All of the responses are marked in green.

Major comments:

POINT 1: -How and where data was generated and then used further during the workflows is not always easy to follow. Some details are missing and some references are not the primary literature while at least one simply does not contain the implied statement, e.g. reference 2. Thus, the manuscript requires sharpening and some rewriting.

RESPONSE 1: We thank the reviewer for the suggestions. We have added details to Supplementary Table 1 wherein it describes the origin of datasets and the mass spectrometry instruments used. The Discovery and Verification workflows consist of the same software tools for all datasets however the parameters vary depending on the datasets. We have customized the software parameters to the nature of the data generated and used methods described in the associated publications. We mention these dataset-specific parameters in Supplementary Table 1. We have also replaced reference 2 with an appropriate reference (Dyer, O. Covid-19: Variants Are Spreading in Countries with Low Vaccination Rates. BMJ 2021, 373, n1359, doi:10.1136/BMJ.N1359).

POINT 2: The authors tried to establish two bioinformatics workflows to identify strain-specific MS-detectable SARS-CoV-2 peptides. From a list of 906 unique peptides, six strain specific, high quality peptides were identified. While in general this workflow can be useful to identify MS-detectable peptides after initial analysis, it is still very limited to the knowledge of a specific strain. Furthermore, the authors claim usefulness for monitoring, vaccine and therapeutics development, without discussing any of the limitations of MS with respect to existing techniques. The set goal of reliably identifying strain specific peptides was accomplished but should be discussed as what it is, the identification of strain-specific peptides using MS (see also below). 

RESPONSE 2: We would like to thank the reviewer for the comment. We have incorporated changes regarding vaccine development and therapeutics in the Discussion section, noting that our work is most relevant to diagnostic applications, with the added value of detecting some proteins that may act as treatment targets (lines 385-390,409-416,423-426 in the revised manuscript). We agree that this method is not designed to discover new strains a priori, but to identify and rigorously confirm the presence of known strain-specific sequences that can be useful for detection and monitoring.

POINT 3: l-80- Which sequences were used and where were they taken from?

RESPONSE 3:Thank you to the reviewer for this question. We have added the source of our information in the Materials and Methods section (lines 131-133 in the revised manuscript).

POINT 4: -l. 103 How were these datasets selected? This is crucial, since the detected peptides depend on this, in the first as well as the second round. Along those lines, in l. 105-108 why are some datasets listed but others were not? In l. 105-108, more background for these sets are needed (single samples pooled?).

RESPONSE 4: Thank you to the reviewer for the comment. We have incorporated additional information regarding these datasets in Supplementary Table 1 (which complements the information in Figure 1), and further helps to explain their selection for our study. These datasets were chosen as they were publicly available on the ProteomeXchange Data repository, collected during different time points during the pandemic directly from patients in the clinic, and obtained from varied geographical areas. With this diverse set of datasets, we decided to investigate them for the detection of strain-specific peptides, as these datasets would represent a cross-section of samples collected during different stages of the pandemic and in different locations.  Also, the dataset PXD034582 (mentioned in lines 128-129 in the revised manuscript), consists of three different datasets collected during August 2021, September 2021, and January 2022, and therefore are listed differentially.  Each of the raw files consisted of pooled patient samples. 

POINT 5: -l. 167 Define peptide reports used, MQ does not generate a peptide report. 

RESPONSE 5: We thank the reviewer for the comment. We have made the required corrections. MaxQuant does generate a peptides.txt file as an output which was used during our analysis (lines 192,241-242 in the revised manuscript).

POINT 6: -l. 173 List of 17 previously described peptides. Described where?

RESPONSE 6: Thank you reviewer for your comment. We have incorporated the clarification regarding the 17 peptides in the 2.3 subsection (lines 242-244 in the revised manuscript). Our BLAST-P and Peptide report analysis showed that there were 17 peptides that were specific to the PANGO lineage.

POINT 7: -l. 199 “in silico translated protein sequences” – overall this makes sense but in this context a bit confusing, since the work is based on empirical data. Please clarify.

RESPONSE 7: Thanks to the reviewer for this comment. We have added an additional reference regarding the in-silico peptide sequences (line 306, references 21 and 28 in the revised manuscript).  We used this in a prior study to determine viral peptides that could be detected in MS data from clinical samples.

POINT 8: -l. 216 Define quality (in spectral quality), how was that assessed. It would be useful and supportive to add additional example spectra (rejected, included/assigned) to the supplement. See also Fig. 3 - Omicron: here the 3x3 rule is not met since only one consecutive b-ion despite many y-ions. This seems to be a high confidence peptide. Please clarify when to break the 3x3 rule.

RESPONSE 8: Thanks to the reviewer for this comment. As per the reviewer’s suggestion, we have added Supplementary figure 2 which shows the manual inspection of the spectra. We have clarified that the accepted spectra could have either consecutive b or consecutive y ions, but having three consecutive b AND three consecutive y ions was not a requirement (Line 234, SupplementaryData 1- Figure 2 in the revised manuscript).

POINT 9: -l. 273-276 The existing datasets were processed using a defined sequence. In this case it is not possible to define new, previously unknown peptides.

RESPONSE 9: Thank you for the reviewer's comment. We have made the changes in the results section that clarifies that the Discovery workflow doesn’t detect “new” peptides, but rather identifies peptides that are present in the sequence database and might represent sequences specific to viral variants (lines 189-194,220-226 in the revised manuscript). Our discovery workflow determines which peptides can be detected followed by the verification workflow to rigorously evaluate the veracity of these detected sequences, using additional screening, confidence, and quality metrics.

POINT 10: -l. 295-300 Monitoring peptides directly is desirable, however the usefulness for vaccine and therapeutic development remains limited. The limitations of the approach need to be discussed. For example: MS can (with little exceptions) only detect structural viral proteins. No changes in the polymerase or nsp3, which are targets for antivirals and the latter has an elevated mutation rate compared to other non-structural proteins, could be detected.

RESPONSE 10: The reviewer raised an interesting issue. We acknowledge that direct detection of viral proteins with mutated sequences can help in determining targets for therapies, but there are limitations. We have modified the discussion (see lines 385-390,409-416,423-426 in the revised manuscript) to mention that there is some value in direct protein detection related to vaccines/therapies, but the primary value of this work is for diagnostic applications. We have removed text in the Discussion for the initial manuscript that discussed the vaccine/therapy value in more detail. It should be noted that in our previous work, we detected a few non-structural proteins from the cell culture datasets (Reference 14).

MINOR:

POINT 1: -Throughout the text some words were used very repetitively mainly rigorous and rigorously. On most occasions, this word has no added value and could be removed.

RESPONSE 1: The suggested changes have been made.

POINT 2: -l. 15-16 Repetition of caused

RESPONSE 2: The suggested changes have been made (line 16).

POINT 3: -l. 38 Repetition of severe

RESPONSE 3: The suggested changes have been made(line 39).

POINT 4: -l. 41 wrong source -DONE

RESPONSE 4: Thank you to the reviewer for pointing this out. We have made the necessary change to the citation (line 41; Reference 2; Dyer, O. Covid-19: Variants Are Spreading in Countries with Low Vaccination Rates. BMJ 2021, 373, n1359, doi:10.1136/BMJ.N1359).

POINT 5: -l. 43 not the primary source

RESPONSE 5: Thank you, we have made the suggested change to the citation (line 46, Reference 3; Wu Y, Kang L, Guo Z, Liu J, Liu M, Liang W. Incubation Period of COVID-19 Caused by Unique SARS-CoV-2 Strains: A Systematic Review and Meta-analysis. JAMA Netw Open. 2022;5(8):e2228008. doi:10.1001/jamanetworkopen.2022.28008).

POINT6: -l. 78 “cover major continents” has no meaning, just 4 continents were covered. Why are certain continents considered major and others minor.

RESPONSE 6: Thank you for the reviewer’s comment. We have reworded this sentence (line 84). 

“These datasets cover a timeline starting from March 2020 to January 2022 and cover 7 countries and three continents (Figure 1, Supplementary Table 1 )”

POINT 7: -Fig. 1 - PXD020394 was a nasal swab not a nasopharyngeal swab according to ProteomeXchange and PXD034582 is not available yet

RESPONSE 7: Thank you to the reviewer for pointing this out. We have made the required changes. PXD034582 will be made available after publication (Figure 1, line 129 in the revised manuscript).

POINT 8: -Table 1 First line very thick use the width of the page

RESPONSE 8: Table 1 has been formatted.

POINT 9: -l. 240 Seems redundant

RESPONSE 9: The suggested change was made (line 344, deleted lines in the revised manuscript).

POINT 10: -l. 263 Period missing

RESPONSE 10: The suggested change was made (line 368 in the revised manuscript).

POINT 11: -l. 284 It is not clear if it is 906 peptides including strain-specific or 912 peptides. Rephrase. 

RESPONSE 11: Thank you for the reviewer’s comment. We have re-phrased this sentence (line 411 in the revised manuscript). 

POINT 12: -l. 302 change is to in

RESPONSE 12: We have made the suggested change (line 428 in the revised manuscript).

Reviewer 3 Report

Comments to Authors:

In this manuscript, the authors used a bioinformatics workflow that allows the identification of a set of SARS-CoV-2 variant peptides which could be used to develop further MS-based methods to monitor the status of the SARS-CoV-2 landscape as the pandemic progresses.

Overall, the manuscript is well structured, and the bioinformatics analysis performed here is very solid and provides beneficial information.

Here I have some comments and suggestions. Hopefully, these suggestions will improve the quality and increase the strength of the manuscript.

Together with the fragmentation (MS/MS) data, If authors could include the MS peak (chromatogram) of each peptide that will be helpful to understand the signal abundance of the peptide.

Alternatively, if authors could collect/provide any kinds of quantitative data (such as PSM data) of each variant peptide that would also be helpful to understand the abundance of that particular peptide for selecting further MS-based assay (PRM/MRM/SRM).

Table 1: Column (Peptide info). Please include the WT sequences together with the variant-specific sequences.

Author Response

We would like to thank the reviewer for their comments. All the responses are marked in green.

POINT 1: Together with the fragmentation (MS/MS) data, If authors could include the MS peak (chromatogram) of each peptide that will be helpful to understand the signal abundance of the peptide.

RESPONSE 1: Thanks to the reviewer for their comment. We have added the intensities of the verified variant peptide (if available) in Supplementary Table 4.

POINT 2: Alternatively, if authors could collect/provide any kinds of quantitative data (such as PSM data) of each variant peptide that would also be helpful to understand the abundance of that particular peptide for selecting further MS-based assay (PRM/MRM/SRM).

RESPONSE 2: Thanks to the reviewer for their comment. In order to address this, we have added Supplementary Table 4, in which we have mentioned the number of PSMs that are observed after the PepQuery analysis with a P-value of 0.05. This is an indication that these peptides are strong enough in their signal to be detected by at least one of the two search algorithms used in the discovery workflow and the tool PepQuery in the verification workflow. 

POINT 3: Table 1: Column (Peptide info). Please include the WT sequences together with the variant-specific sequences.

RESPONSE 3: We appreciate the reviewer’s comment and have added the WT sequences as an additional column to Table 1 (line 327 in the revised manuscript).

Round 2

Reviewer 2 Report

The authors have addressed our comments satisfactorily. Only a few minor points need to be addressed prior to publication.

Major comments:

The paper overall is still hard to read and may benefit from additional early, clear explanation of what was achieved in the end. Suggestion: rewrite the two last sentences of the abstract to make clear that in the end six strain specific peptides suitable for confident detection via mass spectrometry were identified with the proposed workflow.

Minor:

-52 Change detects to identifies? Or restructure sentence to better reflect what was implied

-101 potential studies

-104 repetition of clinical

-112 The authors commented in the answers to the last review, that all chosen samples were pooled samples. This information ought to be provided directly in the manuscript.

-127-132 split into two sentences and rephrase, content is hard to grasp.

-179-180 repetition of using

-188 as belonging to

-230-237 End of first paragraph and beginning of second paragraph are redundant rephrase.

-265-271 Move table description completely above table

- rigorous and rigorously are used less often than before but are still very frequently used

Author Response

We would like to thank the reviewer for their comments. We have incorporated all the required changes (below in green).

Major comments:

POINT 1: The paper overall is still hard to read and may benefit from additional early, clear explanation of what was achieved in the end. Suggestion: rewrite the two last sentences of the abstract to make clear that in the end six strain-specific peptides suitable for confident detection via mass spectrometry were identified with the proposed workflow.

RESPONSE: We would like to thank the reviewer for this comment. We have added the suggested change to the summary part in the discussion and conclusion section. (See Lines 25-28,78-79, 344-346).

Minor:

POINT 1:-52 Change detects to identifies? Or restructure sentence to better reflect what was implied

RESPONSE: Thanks to the reviewer for the comment. The suggested change has been made (line 49).

POINT 2:-101 potential studies

RESPONSE: The suggested change has been made (Please see Line 103).

POINT 3:-104 repetition of clinical

RESPONSE: The suggested change has been made (line 105).

POINT 4:-112 The authors commented in the answers to the last review, that all chosen samples were pooled samples. This information ought to be provided directly in the manuscript.

RESPONSE: Thanks to the reviewer for the comment. The suggested change has been made (lines 121-123) “Note that the datasets collected in Sao Paulo (PXD021328 and PXD034583) were pooled clinical samples, wherein each raw file contained data from two patients.”

POINT 5:-127-132 split into two sentences and rephrase, content is hard to grasp.

RESPONSE: The suggested changes have been made (see lines 119-121).

POINT 6:-179-180 repetition of using

RESPONSE: The suggested changes have been made. (see line 172).

POINT 7:-188 as belonging to

RESPONSE: The suggested changes have been made (see line 192).

POINT 8:-230-237 End of first paragraph and beginning of second paragraph are redundant rephrase.

RESPONSE: We thank the reviewer for this comment. We reworded the end of the first paragraph and the beginning of the next paragraph. (See lines 236-241).

POINT 9:-265-271 Move table description completely above table

RESPONSE: The description of Table 1 was moved to the top of the table. (Lines 270-276).

POINT 10:- rigorous and rigorously are used less often than before but are still very frequently used

RESPONSE: The suggested changes have been made.(see lines 93,281,314).